# How does birth order influence full immunization coverage among children aged 12–23 months in India? evidence from the National Family Health Survey

Tanmoy Ghosh[1]*, Puja Das[2], Apurba Sarkar[3], Pradip Chouhan[4]*

1 Department of Geography, University of Gour Banga, Malda, West Bengal, India, 2 Department of Geography, University of Gour Banga, Malda, West Bengal, India, 3 New Integrated Government School (NIGS), Habibpur, West Bengal, Sub-Ordinate Education Service (Group-A), School Education Department, Government of West Bengal, West Bengal, India, 4 Department of Geography, University of Gour Banga, Malda, West Bengal, India

* tanmoy.ghosh202020@gmail.com (TG), pradipchouhanmalda@gmail.com (PC)

## Abstract

Despite the availability of free basic vaccination programmes, the disparity continues in immunization coverage among children aged 12–23 months in India, particularly with increasing birth orders. Using the data from the NFHS 5 (2019–21), the study seeks to investigate how birth order affects the likelihood of children's immunization aged 12–23 months in India. The analysis included a nationally representative sample of 43,436 children aged 12–23 months. Bivariate LISA and multilevel logistic regression models were performed to assess spatial and statistical patterns. The study found that spatial autocorrelation analysis indicated a positive association between 1st and 2nd birth orders and child full immunization coverage. However, the likelihood of full immunization declines significantly with increasing birth order [2nd (OR: 0.91); 3rd (OR: 0.74); 4th (OR: 0.64); 5th (OR: 0.56); 6 and more (OR: 0.41)]. Similarly, other socio-demographic covariates such as female children [AOR: 0.95], children who belong to Muslim families [AOR: 0.72], and resided in the northeastern part [AOR: 0.52] of India had a lower likelihood of being completely immunized. Therefore, the present study underscores the urgent need for targeted public health interventions that address both familial and structural barriers to immunization, specifically for mothers who have more than two children, to ensure that no child is left behind, regardless of their birth position within the family.

## Introduction

Child immunization is not just a cost-effective intervention against vaccine-preventable diseases in reducing child morbidity and mortality, but it also significantly improves child health as well as the economic development of a country. According to

**Data availability statement:** The study is based on secondary data which is available in the public domain https://dhsprogram.com/data/available-datasets.cfm

**Funding:** The author(s) received no specific funding for this work.

**Competing interests:** The authors have declared that no competing interests exist.

the World Health Organization, Immunization currently prevents 3.5 million to 5 million deaths every year from vaccine-preventable diseases [1]. Child full immunization coverage ensures that children who received all recommended vaccines, that is three doses of diphtheria-tetanus-pertussis (DPT) vaccine, three doses of polio vaccine, one dose of Bacillus Calmette-Guerin (BCG) vaccine, and one dose of measles vaccine [2] within the first year of life, have been a critical priority in global and national health agendas [3]. In India, the Expanded Programme on Immunization (1978), renamed the Universal Immunization Programme (1985), has completed almost 30 years of journey; still, significant disparities remain across the regions influenced by a variety of socioeconomic, demographic, and cultural factors [4,5].

One less-explored but potentially one of the most important determining factors of child immunization coverage is birth order, a demographic characteristic that refers to the sequence of a child's birth within a family. Previous research suggests that first-born children are more likely to receive complete immunization than later-born siblings [6–8]. An explanation for this would be that lower-birth-order children tend to receive more focused care and parental attention, as families may have more resources and time to bring them up. In contrast, children with higher birth order may face reduced parental attention and limited resource allocation because of the competing demands of large family sizes. Parental attention, which plays an important role in immunizing children, may be of various kinds, including household financial resources, women's time available for children, parental characteristics, and the specific characteristics of children, including birth order [9]. Recent studies on health and education indicate an inverse relationship between household size or birth order and disparities in health and education outcomes [10–13].

Although India's Immunization programmes have made considerable progress, the persistent gap linked to birth order remains a significant public health concern. While many studies have reported an inverse relationship between birth order and immunization coverage [14–17]. However, the mechanisms and pathways are not well explored. In particular, there is as such no Indian studies which have addressed critical interplay between the birth order and immunization status through a spatial lens. More causal analyses are needed, specifically studies that focus on maternal birth order, particularly studies that adjust for different controlling factors such as child level, maternal and household-level, health care and community-level characteristics. Therefore, the present study attempts to determine how birth order influences full immunization coverage among children aged 12–23 months in India. Furthermore, it also explores the geographic distribution and clustering patterns of birth order and child full immunization coverage across regions. By analyzing the pattern of different vaccine uptake across different birth orders and other selected socio-demographic and household characteristics, this study aimed to provide insights into potential disparities and inform targeted interventions to improve immunization rates among all children. The findings are expected to support strategies that promote universal immunization and accelerate progress toward achieving Sustainable Development Goal (SDG) 3, which focuses on ensuring good health and well-being for all.

## Materials and methods

### Study design and sample

To explore the association between birth order and child full immunization coverage, data were collected from the fifth round of the National Family Health Survey [18] conducted in 2019−2021. A nationally representative cross-sectional survey was conducted under the supervision of the Ministry of Health and Family Welfare (MoHFW) of India, with technical coordination and guidance from the International Institute of Population Science (IIPS), Mumbai. It provides detailed information on several key topics such as health and family welfare, fertility levels, infant and child mortality, maternal and child health, and covers other relevant indicators at both the national and state levels. Stratified random sampling techniques were used to collect data from the national, state/union territory, and district levels. A total of 724,115 women aged 15−49 years and 101,839 men aged 15−54 years were collected from 636,699 households with response rates of 97% and 93%, respectively (NFHS-5).

### Study participants

First, the study included 232,920 children aged 0–59 months who were born in the last five years. Second, 1,89,484 children were excluded from the analysis; among them, 8,702 children died, and 180,782 children were below 12 months or above 23 months of age. Finally, statistical analysis of this study was performed on 43,426 children from 12–23 months of age (**Fig 1**).

### Ethical approval

Since this study is based on secondary data, which is available in the public domain (https://dhsprogram.com/data/available-datasets.cfm), no ethical approval is required.

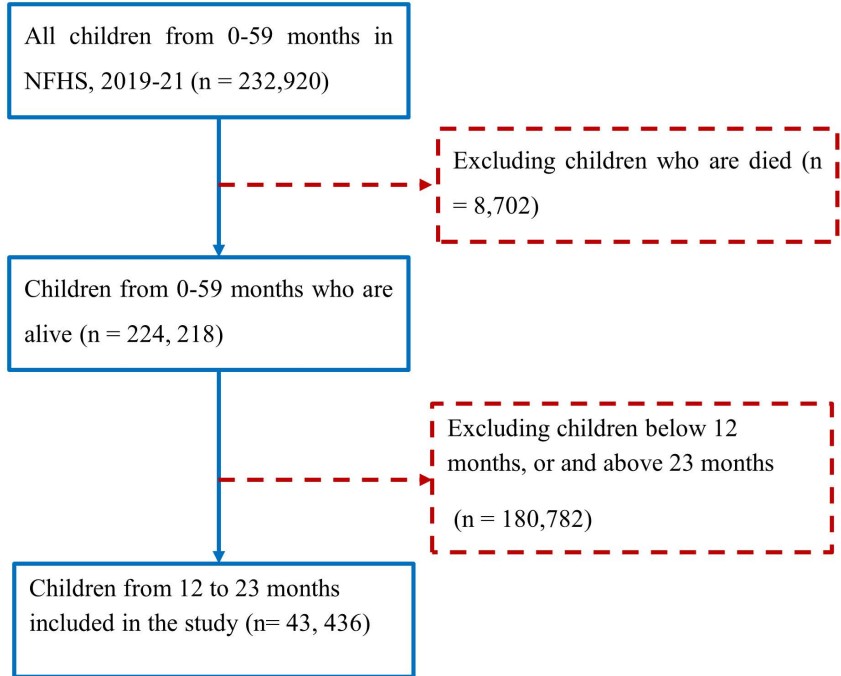

**Fig 1. Flowchart depicting the inclusion of children aged 12-23 months in the analysis from NFHS-5 (2019-2021), India.**

## Dependent variable

To assess the influence of birth order on full immunization coverage, the outcome variable of the study was full immunization of children aged 12–23 months. As per the WHO recommendations, children should have received basic primary scheduled vaccines (BCG, three doses of DPT, 3 doses of Polio and one dose of measles) at the age of 12 months, while the period of 23 months allows for some delay in vaccine administration [19]. Children who received all the basic vaccines, that is, one dose of BCG (Bacillus, Calmette-Guerin), three doses of DPT (Diphtheria Pertussis Tetanus), three doses of Polio, and one dose of measles vaccine at any time before the survey were considered for full vaccination. Particular vaccination data were collected based on information provided by the mothers or vaccination cards only. Children who received the basic vaccine were coded 1 (yes) or 0 (no).

## Independent variable

The key independent variable in this study is birth order, categorized into six groups: 1st, 2nd, 3rd, 4th, 5th, and ≥ 6th). The control variables encompass a range of child, maternal and household-level, healthcare, and community-level characteristics. Child-related variables include sex of the child (male and female) and child size at birth (large, average, and small). Maternal characteristics included maternal age (15–24, 25–34, and 35+), maternal education (no education, primary, secondary, and higher), current marital status (no and yes), exposure to mass media (no, partial and full), sex of the household head (male and female), wealth index (poorest, poorer, middle, richer and richest), religion (Hinduism, Islam, Christian, and others), and social category (SC, ST, OBC and others). Healthcare-related variables include four or more ANC visits (no and yes), and place of delivery (home, public, and private). Lastly, community-level variables comprised place of residence (urban and rural) and region (North, Central, East, Northeast, West, and South).

## Statistical analysis

A multilevel binary logistics regression model was used to identify the association between birth order and full immunization coverage. First, we calculated the proportion of complete and incomplete immunization coverage, and then calculated the weighted percentage distribution to examine the selected variables. Bivariate percentages were also calculated to examine the variation in birth order and other socio-demographic and household characteristics in relation to full child immunization coverage. Subsequently, Pearson's chi-squared test was conducted to assess the statistical significance of these variables.

The study also applied spatial analytical techniques to examine the association between birth order and full child immunization coverage using the GeoDa software. Bivariate Moran's I statistics (LISA), along with a spatial significance model, were utilized to draw a spatial conclusion. This investigation utilized the district level as the spatial unit of analysis, covering 707 districts throughout India. Bivariate Moran's I was selected to measure the spatial correlation between two distinct variables: the prevalence of a specific birth order in one district and the full immunization coverage in adjacent districts. This method effectively identifies spatial associations, aligning with our primary objective, as opposed to modeling predictive relationships, for which Geographically Weighted Regression (GWR) would be more appropriate. The bivariate approach, in contrast to a standard univariate Local Moran's I that identifies clusters of a single variable, was crucial for directly examining the spatial relationship between the independent and dependent variables. The bivariate version of Moran's I is represented as follows:

$$Bivariate\ Moran's\ I = \frac{n}{S_0} \times \frac{\Sigma_i \Sigma_j W_{ij}\,(x_i - \overline{X})(Y_j - \overline{Y})}{\Sigma_i\,(y_i - \overline{Y})^2}$$

Where x and y are the variables being analyzed; $\overline{X}$ and $\overline{Y}$ represent the means of x and y, respectively; n is the total number of spatial units; $W_{ij}$ is the standard weight matrix, which defines the relationship or connectivity between spatial units i and j, with zeroes on the diagonal; and $S_0$ is the sum of all spatial weights, given by $S_0 = \Sigma_i \Sigma_j W_{ij}$.

In the final stage of our analysis, a multilevel binary logistic regression model was used to identify the association between birth order and full immunization coverage of children, while adjusting for other sociodemographic and household covariates. A multilevel model was essential to account for the hierarchical structure of the NFHS data, where children (level 1) are nested within communities, or **Primary Sampling Units (PSUs, level 2)**. The PSU, which is a village or a census block, was chosen as the cluster level because individuals within the same PSU share community-level characteristics and are not statistically independent, thus violating the assumptions of standard regression. This approach corrects for clustering and provides more accurate estimates. The analysis employed a random intercept model, which assumes that the baseline odds of immunization vary across communities (PSUs), while the effects of the independent variables are consistent. Seven models (0–6) were developed. Model 0 served as the empty model, and it denotes the variance in immunization coverage attributed to the primary sampling unit (PSU). Model 1 was fitted to contain only birth order; Model 2–5 progressively added child, maternal & household-level, healthcare, and community characteristics; and the final model (Model 6) included birth order along with all incorporated covariates. The results are presented using the crude odds ratio (cOR) and adjusted odds ratio (aOR), along with their respective 95% confidence intervals. Statistical significance was set at $p < 0.01$, $p < 0.05$, and $p < 0.1$, respectively. All seven models included both fixed and random effects models. The random effects show the variation in full immunization coverage attributed to the primary sampling unit (PSU), measured using their Intra-Class Correlation (ICC). In contrast, the fixed effects model estimated the association between birth order and full child immunization, along with other covariates. Model fit and comparisons were assessed using Akaike's Information Criterion (AIC), where a lower AIC value indicates a better-fitting model. All statistical analyses were performed using STATA (version 17), whereas spatial analysis was conducted using GeoDa (version 1.22).

## Result

Table 1 presents the sociodemographic and household characteristics of children aged 12–23 months and their mothers. Approximately 77.08% of the total sample population achieved full immunization coverage. The majority of children in the sample were in the first to second birth order. Among the studied children, 52% and 48% were male and female, respectively, with approximately 70% of the children having an average size at birth. Most of the women were aged 25–34 years, and nearly one-fourth (18.98%) had no educational attainment. Over 99% of the women were currently married, while only 1.36% were fully exposed to mass media. The majority of household heads were male (85.04%). A large proportion of the children belonged to the poorest (23.90%) and poorer (21.36%) wealth quintiles, came from Hindu families (79.56%), and were in the OBC category (45.78%). More than 58% of the women had four or more ANC visits, and nearly two-thirds of the mothers (63.06%) delivered their children to public health institutions. The majority of the children resided in rural areas (73.10%) and were from the central (27.63%) and eastern (26.55%) regions of India.

Table 2 illustrates the analysis of full immunization coverage among children aged 12–23 months by birth order and various socio-demographic and household characteristics. With increasing birth order, full immunization coverage decreases significantly. Full immunization coverage was significantly higher among male children (77.50%) and among those with an average size at birth (77.85%). Women aged 15–24 years (77.40%), who completed secondary (79.57%) or higher education (79.06%), were currently married (77.15%), and were partially exposed to mass media (79.44%). Their children were fully immunized, and this association was statistically significant. Male-dominated (77.16%) households had higher rates of fully immunized children than female-dominated (76.64%) households; however, this association was not statistically significant. The rate of complete immunization coverage was significantly higher among children from the middle (79.98%), richer (79.71%), or richest (79.20%) wealth quintiles; those belonging to other religions (80.73%) were in OBC categories (77.55%); and those from the southern (82.57%) regions of India. Mothers who visited at least four or more ANC (81.48%) and delivered their children in public health facilities (78.96%) had their children significantly more fully immunized. Children who resided in rural areas had higher rates of full immunization; however, this association was not statistically significant.

**Table 1. Sociodemographic and household characteristics of women with children aged 12−23 months in India, 2019−21.**

| Characteristics | Frequency | Weight % |
|---|---|---|
| Child Full Immunization | | |
| No | 9,852 | 22.92 |
| Yes | 33,221 | 77.08 |
| *Child Characteristics* | | |
| **Birth Order** | | |
| 1 | 16,941 | 39.71 |
| 2 | 14,463 | 34.33 |
| 3 | 6,654 | 14.89 |
| 4 | 2,947 | 6.35 |
| 5 | 1,326 | 2.65 |
| 6 or more | 1,105 | 2.08 |
| **Sex of the child** | | |
| Male | 22,537 | 52.00 |
| Female | 20,899 | 48.00 |
| **Child size at birth** | | |
| Large | 8,087 | 19.51 |
| Average | 30,511 | 69.70 |
| Small | 4,384 | 10.79 |
| *Maternal & Household-level characteristics* | | |
| **Maternal age** | | |
| 15-24 | 16,847 | 40.84 |
| 25-34 | 23,521 | 53.30 |
| 35+ | 3,068 | 5.86 |
| **Maternal education** | | |
| No education | 8,475 | 18.98 |
| Primary | 5,171 | 11.33 |
| Secondary | 23,209 | 52.45 |
| Higher | 6,581 | 17.24 |
| **Currently married** | | |
| No | 526 | 0.81 |
| Yes | 42,436 | 99.19 |
| **Exposure to mass media** | | |
| No | 12,557 | 28.24 |
| Partial | 30,322 | 70.40 |
| Full | 557 | 1.36 |
| **Sex of the household head** | | |
| Male | 36,867 | 85.04 |
| Female | 6,569 | 14.96 |
| **Wealth Index** | | |
| Poorest | 11,386 | 23.90 |
| Poorer | 10,073 | 21.36 |
| Middle | 8,518 | 19.86 |
| Richer | 7,446 | 18.75 |
| Richest | 6,013 | 16.13 |

*(Continued)*

**Table 1.** (Continued)

| Characteristics | Frequency | Weight % |
|---|---|---|
| **Religion** | | |
| Hinduism | 32,212 | 79.56 |
| Islam | 6,106 | 16.13 |
| Christian | 3,340 | 1.98 |
| Others | 1,778 | 2.33 |
| **Social category** | | |
| SC | 8,952 | 24.44 |
| ST | 8,548 | 10.57 |
| OBC | 16,705 | 45.78 |
| Others | 6,978 | 19.21 |
| *Healthcare characteristics* | | |
| **Four or more ANC visit** | | |
| No | 16,930 | 41.07 |
| Yes | 23,394 | 58.93 |
| **Place of delivery** | | |
| Home | 4,937 | 9.25 |
| Public | 28,623 | 63.06 |
| Private | 9,876 | 27.69 |
| *Community characteristics* | | |
| **Place of residence** | | |
| Urban | 8,879 | 26.90 |
| Rural | 34,557 | 73.10 |
| **Region** | | |
| North | 7,869 | 12.97 |
| Central | 11,266 | 27.63 |
| East | 8,574 | 26.55 |
| Northeast | 6,189 | 3.54 |
| West | 3,895 | 12.41 |
| South | 5,643 | 16.91 |

A significant spatial dependency was found between birth order and full child immunization coverage throughout India. Fig 2 displays the bivariate LISA maps, significance maps, and Moran scatter plots illustrating the clustering of birth order and full child immunization coverage across 707 districts of India. Maps 2a and 2b show the Bivariate LISA cluster map, which shows that around 87 and 76 districts comprised hotspots concerning the higher rates of full immunization coverage among 1st and 2nd birth-order children, respectively. The Moran Scatter plot indicates a positive relation between 1st & 2nd birth order and full child immunization coverage. In comparison, 78 and 73 districts were identified as cold spots, showing a lower proportion of full immunization coverage among 1st and 2nd born children. Most cold spots were located in Bihar, Uttar Pradesh, and some North East states of India.

Similarly, maps 2c, 2d, 2e, and 2f display that 38, 23, 24, and 17 districts were included in the hotspot zones concerning the rate of full immunization coverage among 3rd to 6th or more birth order children. A noticeable trend is that as the birth order increases (3rd or more), the number of hotspot zones for full child immunization coverage decreases across the districts in India. The Moran scatter plot further illustrates a negative relationship between birth order and full child immunization coverage.

**Table 2. Child full immunization coverage by birth order and other socio-demographic and household characteristics of women with children aged 12−23 months in India, 2019−21.**

| Variables | Full Immunization Coverage | | $x^2$ | p value |
|---|---|---|---|---|
| | **No** | **Yes** | | |
| *Child Characteristics* | | | | |
| **Birth Order** | | | 311.5973 | <0.001 |
| 1 | 20.27 | 79.73 | | |
| 2 | 22.31 | 77.69 | | |
| 3 | 25.84 | 74.16 | | |
| 4 | 28.35 | 71.65 | | |
| 5 | 30.64 | 69.36 | | |
| 6 or more | 36.23 | 63.77 | | |
| **Sex of the child** | | | 5.9378 | 0.015 |
| Male | 22.50 | 77.50 | | |
| Female | 23.37 | 76.63 | | |
| **Child size at birth** | | | 27.8768 | <0.001 |
| Large | 24.10 | 75.90 | | |
| Average | 22.15 | 77.85 | | |
| Small | 24.01 | 75.99 | | |
| *Maternal & Household-level characteristics* | | | | |
| **Maternal age** | | | 17.0071 | <0.001 |
| 15-24 | 22.60 | 77.40 | | |
| 25-34 | 22.78 | 77.22 | | |
| 35+ | 26.31 | 73.63 | | |
| **Maternal education** | | | 416.2793 | <0.001 |
| No education | 30.94 | 69.06 | | |
| Primary | 24.06 | 75.94 | | |
| Secondary | 20.43 | 79.57 | | |
| Higher | 20.94 | 79.06 | | |
| **Currently married** | | | 19.5215 | <0.001 |
| No | 31.30 | 68.70 | | |
| Yes | 22.85 | 77.15 | | |
| **Exposure to mass media** | | | 454.7804 | <0.001 |
| No | 28.75 | 71.25 | | |
| Partial | 20.56 | 79.44 | | |
| Full | 24.45 | 75.55 | | |
| **Sex of the household head** | | | 1.9360 | 0.164 |
| Male | 22.84 | 77.16 | | |
| Female | 23.36 | 76.64 | | |
| **Wealth Index** | | | | |
| Poorest | 28.03 | 71.97 | 402.7644 | <0.001 |
| Poorer | 23.84 | 76.16 | | |
| Middle | 20.02 | 79.98 | | |
| Richer | 20.29 | 79.71 | | |
| Richest | 20.80 | 79.20 | | |
| **Religion** | | | 295.7049 | <0.001 |
| Hinduism | 21.97 | 78.03 | | |

*(Continued)*

**Table 2.** (Continued)

| Variables | Full Immunization Coverage | | $x^2$ | p value |
|---|---|---|---|---|
| Islam | 28.29 | 71.71 | | |
| Christian | 21.39 | 78.61 | | |
| Others | 19.27 | 80.73 | | |
| **Social category** | | | 50.5116 | <0.001 |
| SC | 22.70 | 77.30 | | |
| ST | 22.90 | 77.10 | | |
| OBC | 22.45 | 77.55 | | |
| Others | 23.38 | 76.62 | | |
| *Healthcare characteristics* | | | | |
| **Four or more ANC visit** | | | 685.5302 | <0.001 |
| No | 28.54 | 71.46 | | |
| Yes | 18.52 | 81.48 | | |
| **Place of delivery** | | | 687.1066 | <0.001 |
| Home | 35.02 | 64.98 | | |
| Public | 21.04 | 78.96 | | |
| Private | 23.17 | 76.83 | | |
| *Community characteristics* | | | | |
| **Place of residence** | | | 0.1045 | 0.747 |
| Urban | 24.00 | 76.00 | | |
| Rural | 22.51 | 77.49 | | |
| **Region** | | | 561.3998 | <0.001 |
| North | 19.88 | 80.12 | | |
| Central | 27.28 | 72.72 | | |
| East | 21.13 | 78.87 | | |
| Northeast | 32.50 | 67.50 | | |
| West | 24.96 | 75.04 | | |
| South | 17.43 | 82.57 | | |

The results of the multilevel logistic regression analysis are presented in **Table 3**. The first model (Model 0) was the empty model, which contained no explanatory variable, revealing that 23% of the variation in full immunization coverage was due to cluster-level differences (ICC = 0.23, p < 0.001). In Model 1, only the birth order covariate was included to examine how birth order influences a child's full immunization coverage. The findings revealed that the odds of full immunization coverage were likely to be less in 2nd (cOR: 0.91), 3rd (cOR: 0.74), 4th (cOR: 0.64), 5th (cOR: 0.56), and 6th or later (cOR: 0.41) birth order children than in 1st birth order children. The PCV for Model 1 indicates that birth order alone explains 4.13% of the variance at the individual level. Selected child characteristics were included in this analysis in Model 2 (PCV = 7.44%). Although the odds of full immunization slightly increase in 4th and 6 or more birth orders. However, the occurrence of full immunization was significantly lower with an increase in birth order compared with the 1st birth order children. The selected maternal and household-level characteristics were included in Model 3, which further improved the explanatory variable (PCV = 11.57%). Despite slight increases, the likelihood of complete immunization coverage significantly decreased with increasing maternal birth order; specifically, after controlling for all covariates, the odds of full immunization were 34% lower for children of the sixth or higher birth order compared to first-born children (aOR: 0.66), thereby implying birth order as a critical determinant.

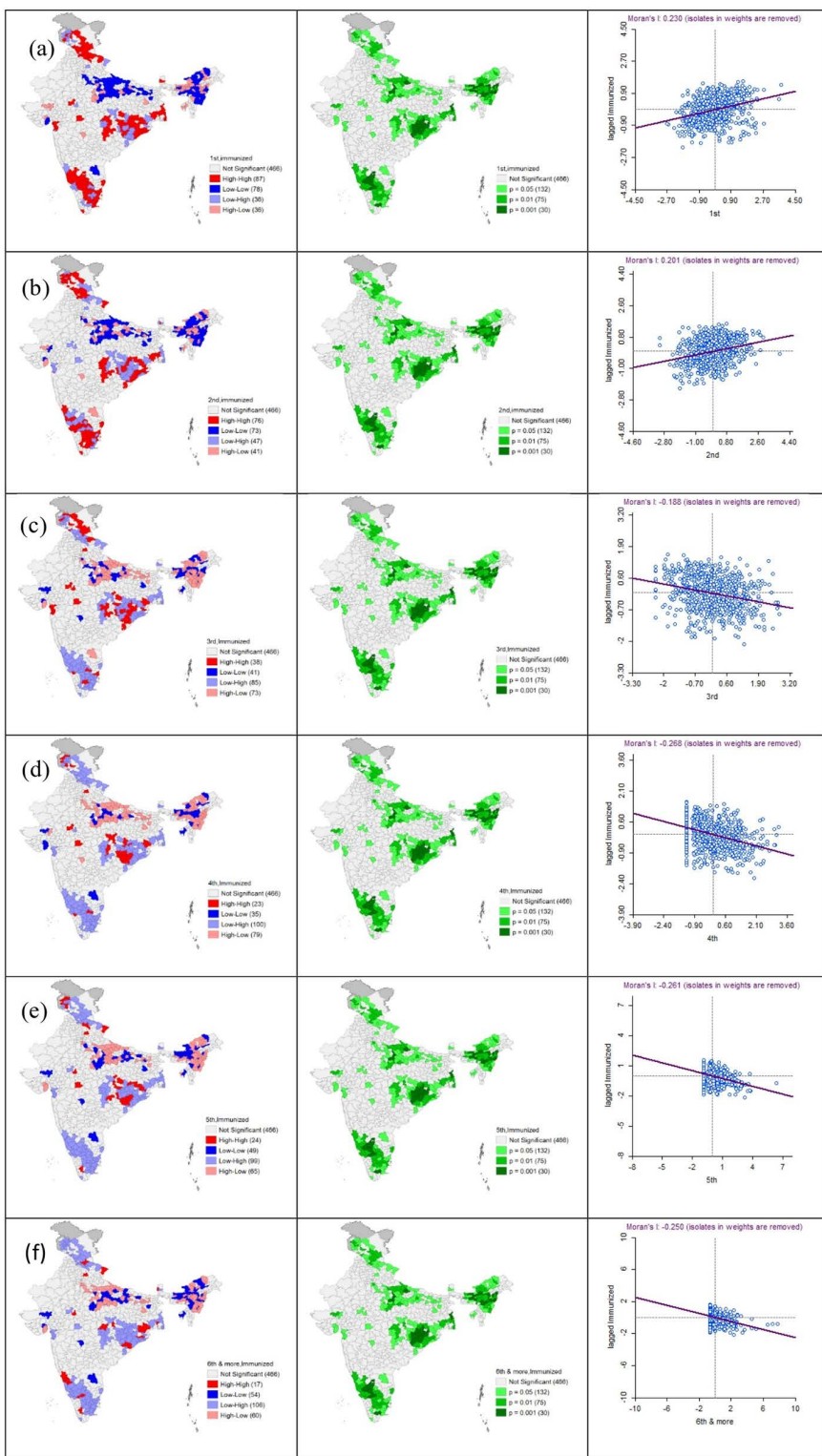

**Fig 2. Bivariate LISA and Significance map showing spatial auto correlation between birth order and child full immunization coverage among children aged 12-23 months in India [(a) first birth order; (b) second birth order; (c) third birth order; (d) fourth birth order; (e) fifth birth order; (f) six and more birth order].**

**Table 3.** Factors associated with child full immunization coverage identified by a multilevel logistic regression model, India, 2019-2021.

| Characteristics | Model 0 | Model 1 | Model 2 | Model 3 | Model 4 | Model 5 | Model 6 |
|---|---|---|---|---|---|---|---|
| | Empty Model | Birth Order (cOR) | Child Characteristics (aOR) | Maternal and Household-level Characteristics (aOR) | Healthcare Characteristics (aOR) | Community Characteristics (aOR) | Full Model (aOR) |
| *Fixed effect results* | | | | | | | |
| *Child Characteristics* | | | | | | | |
| **Birth Order** | | | | | | | |
| 1@ | | 1 | 1 | 1 | 1 | 1 | 1 |
| 2 | | 0.91*** (0.85-0.97) | 0.91*** (0.86-0.97) | 0.89*** (0.83-0.96) | 0.93** (0.87-0.99) | 0.90*** (0.84-0.96) | 0.90*** (0.84-0.96) |
| 3 | | 0.74*** (0.68-0.80) | 0.74*** (0.69-0.80) | 0.78*** (0.72-0.86) | 0.80*** (0.74-0.87) | 0.75*** (0.69-0.81) | 0.79*** (0.72-0.87) |
| 4 | | 0.64*** (0.57-0.71) | 0.65*** (0.59-0.72) | 0.74*** (0.66-0.83) | 0.75*** (0.68-0.84) | 0.67*** (0.60-0.74) | 0.81*** (0.71-0.92) |
| 5 | | 0.56*** (0.48-0.64) | 0.55*** (0.48-0.64) | 0.70*** (0.59-0.82) | 0.71*** (0.61-0.82) | 0.59*** (0.51-0.69) | 0.76*** (0.64-0.90) |
| 6 and more | | 0.41*** (0.35-0.48 | 0.44*** (0.37-0.51) | 0.55*** (0.46-0.66) | 0.56*** (0.47-0.66) | 0.45*** (0.39-0.53) | 0.66*** (0.54-0.79) |
| **Sex of the child** | | | | | | | |
| Male@ | | | 1 | | | | 1 |
| Female | | | 0.93*** (0.88-0.98) | | | | 0.95** (0.89-1.00) |
| **Child size at birth** | | | | | | | |
| Large@ | | | 1 | | | | 1.00 |
| Average | | | 1.16*** (1.08-1.25) | | | | 1.14*** (1.06-1.22) |
| Small | | | 1.01 (0.91-1.12) | | | | 1.03 (0.93-1.15) |
| *Maternal & Household-level characteristics* | | | | | | | |
| **Maternal age** | | | | | | | |
| 15-24@ | | | | 1 | | | 1 |
| 25-34 | | | | 1.15*** (1.07-1.23) | | | 1.16*** (1.09-1.25) |
| 35+ | | | | 1.30*** (1.14-1.47) | | | 1.34*** (1.18-1.53) |
| **Maternal education** | | | | | | | |
| No education@ | | | | 1 | | | 1 |
| Primary | | | | 1.20*** (1.09-1.32) | | | 1.23*** (1.11-1.36) |
| Secondary | | | | 1.38*** (1.27-1.50) | | | 1.38*** (1.27-1.50) |
| Higher | | | | 1.25*** (1.11-1.40) | | | 1.21*** (1.08-1.37) |
| **Currently married** | | | | | | | |
| No@ | | | | 1 | | | 1 |

*(Continued)*

| Characteristics | Model 0 | Model 1 | Model 2 | Model 3 | Model 4 | Model 5 | Model 6 |
|---|---|---|---|---|---|---|---|
| Yes | | | | 1.51*** (1.20-1.90) | | | 1.41*** (1.11-1.80) |
| **Exposure to mass media** | | | | | | | |
| No@ | | | | 1 | | | 1 |
| Partial | | | | 1.33*** (1.24-1.43) | | | 1.28*** (1.19-1.37) |
| Full | | | | 1.03 (0.80-1.33) | | | 0.97 (0.75-1.25) |
| **Sex of the household head** | | | | | | | |
| Male@ | | | | 1 | | | 1 |
| Female | | | | 1.01 (0.94-1.09) | | | 1.03 (0.95-1.12) |
| **Wealth Index** | | | | | | | |
| Poorest@ | | | | 1 | | | 1 |
| Poorer | | | | 1.13*** (1.05-1.23) | | | 1.09** (1.00-1.19) |
| Middle | | | | 1.30*** (1.18-1.42) | | | 1.23*** (1.12-1.36) |
| Richer | | | | 1.32*** (1.19-1.46) | | | 1.32*** (1.17-1.47) |
| Richest | | | | 1.28*** (1.13-1.44) | | | 1.33*** (1.16-1.52) |
| **Religion** | | | | | | | |
| Hinduism@ | | | | 1 | | | 1 |
| Islam | | | | 0.70*** (0.64-0.77) | | | 0.72*** (0.65-0.79) |
| Christian | | | | 0.52*** (0.46-0.58) | | | 0.82** (0.71-0.95) |
| Others | | | | 0.80*** (0.69-0.92) | | | 0.92 (0.78-1.07) |
| **Social category** | | | | | | | |
| SC@ | | | | 1 | | | 1 |
| ST | | | | 1.08 (0.94-1.09) | | | 1.21*** (1.09-1.34) |
| OBC | | | | 1.01 (0.94-1.09) | | | 1.04 (0.96-1.12) |
| Others | | | | 0.97 (0.88-1.07) | | | 1.00 (0.90-1.10) |
| *Healthcare characteristics* | | | | | | | |
| **Four or more ANC visit** | | | | | | | |
| No@ | | | | | 1 | | 1 |
| Yes | | | | | 1.78*** (1.68-1.88) | | 1.62*** (1.53-1.73) |
| **Place of delivery** | | | | | | | |
| Home | | | | | 1 | | 1 |

*(Continued)*

 

**Table 3.** (Continued)

| Characteristics | Model 0 | Model 1 | Model 2 | Model 3 | Model 4 | Model 5 | Model 6 |
|---|---|---|---|---|---|---|---|
| Public | | | | | 2.03*** (1.86-2.20) | | 1.64*** (1.50-1.80) |
| Private | | | | | 1.69*** (1.53-1.86) | | 1.24*** (1.11-1.39) |
| *Community characteristics* | | | | | | | |
| **Place of residence** | | | | | | | |
| Urban@ | | | | | | 1 | 1 |
| Rural | | | | | | 1.17*** (1.08-1.25) | 1.32*** (1.21-1.44) |
| **Region** | | | | | | | |
| North@ | | | | | | 1 | 1 |
| Central | | | | | | 0.63*** (0.58-0.70) | 0.75*** (0.68-0.83) |
| East | | | | | | 0.89** (0.80-0.98) | 1.20*** (1.07-1.34) |
| Northeast | | | | | | 0.43*** (0.39-0.48) | 0.52*** (0.46-0.60) |
| West | | | | | | 0.78*** (0.69-0.88) | 0.77*** (0.68-0.88) |
| South | | | | | | 1.14** (1.02-1.27) | 1.04 (0.92-1.18) |
| ***Random effect model*** | | | | | | | |
| PSU variance (95% CI) | 1.21 (1.09-1.35) | 1.16 (1.04-1.29) | 1.12 (1.01-1.26) | 1.07 (0.95-1.20) | 0.91 (0.80-1.04) | 1.09 (0.98-1.22) | 0.83 (0.72-0.96) |
| ICC | 0.23 | 0.26 | 0.25 | 0.25 | 0.22 | 0.25 | 0.20 |
| Explained Variation (i.e., PCV in %) | Reference | 4.13 | 7.44 | 11.57 | 24.79 | 9.91 | 31.40 |
| Wald chi-square | Reference | 230.27*** | 236.70*** | 734.26*** | 961.44*** | 631.59*** | 1296.76*** |
| LR test vs. Logistic model: chi2 | 836.95*** | 768.86*** | 709.13*** | 617.35*** | 443.22*** | 707.58*** | 336.05*** |
| Log likelihood | −22743.372 | −22630.55 | −22267.74 | −21053.227 | −20596.767 | −22418.658 | −18843.253 |
| Model fit statistics AIC | 45490.74 | 45275.1 | 44555.48 | 42158.45 | 41213.53 | 44863.32 | 37762.51 |

\*\*\* if p < 0.01, \*\* if p < 0.05, \*if p < 0.1, CI = Confidence interval, @ = Reference category, ANC = antenatal care OR-odds ratio, PSU- Primary sampling unit; VIC- variance partition coefficient; ICC- Intra-community correlation coefficient; AIC- Akaike information criterion

Compared to Model 1, after including healthcare characteristics in Model 4, full immunization coverage slightly improved across all birth orders, with a PCV of 24.79%. After incorporating only the community-level factor into the multi-level models, the results showed that the occurrence of full immunization coverage a little bit increased in the 3rd, 4th, 5th, and 6th or later birth orders compared with the 1st birth order. However, the PCV for Model 5 (9.91%) indicates that the explained variation for birth order and selected community-level characteristics was significantly lower at the individual level compared to Models 2, 3, and 4.

The full model contained all the selected child, maternal, healthcare, and community-level factors (Model 6). The results from Model 6 indicate a significantly lower likelihood of full immunization coverage with the increase in birth order,

female children (aOR: 0.95), and children who belong to Muslim families (aOR: 0.72). The odds of full immunization coverage were significantly higher among children who had an average birth size (aOR: 1.14), whose mothers were aged 35 years or older (aOR: 1.34), had completed secondary education (aOR: 1.38), were currently married (aOR: 1.41), and had been partially exposed to mass media (aOR: 1.28). Children who belonged to the richest wealth quintile (aOR: 1.33), ST categories (aOR: 1.21), were born to those mothers who had visited four or more ANC visits (aOR: 1.62), and delivered in public health facilities (aOR: 1.64) were more likely to be fully immunized. The odds of full immunization were more than one time higher among children residing in rural areas (aOR: 1.32) and those from eastern (aOR: 1.20) regions of India. The PCV of Model 6 indicates that 31.40% of the variation in full child immunization coverage across the individual level could be explained by all child, maternal, healthcare, and community-level characteristics.

The random effects models in Model 0 indicated that 23% of the variation in full immunization coverage in India was attributed to intra-class correlation variation (ICC = 0.23). The variation between clusters increased to 26% (ICC = 0.26) in Model 1, but decreased slightly in Model 2 (ICC = 0.25) and remained the same in Model 3 (ICC = 0.25). The intra-class correlation further declined to 22% (ICC = 0.22) in Model 4 and increased again in Model 5 (ICC = 0.25). The 6th and final model showed that the ICC variation in full child immunization coverage ultimately decreased to 20% (ICC = 0.20).

## Discussion

Numerous global strides, such as the Expanded Program on Immunization (EPI) of the WHO, Gavi, and Global Vaccine Action Plan, have been working diligently to reach indispensable vaccines in poor countries where infection-induced child mortality is common [20–22]. However, universal coverage remains a concern in low and middle-income countries [14,23,24]. For instance, South Asian countries are still grappling with uneven child immunization, particularly with a 77.1% share of full immunization in India, which is even lower than in Bangladesh (88.2%) and Nepal (79.2%) [25]. Against this backdrop, several Indian studies have already been credited with coverage [26], heterogeneity [27,28], and socio-economic inequality [2,29], but the spatial association between maternal birth order and full child immunization coverage remains understudied in the Indian context. Therefore, the present study has significant policy implications for universal full immunization coverage by addressing this gap.

Our study found a significant negative relationship between maternal birth order and full child immunization, which aligns with prior studies [30,31].

In addition, the negative spatial association of 3rd and subsequent birth orders and full child immunization coverage was found to be clustered with hot spots, particularly over the districts of Chhattisgarh and Odisha. This could be ascribed to the prevailing low socio-economic status and poor infrastructural setup of the concerned states [32]. For example, Chhattisgarh and Odisha both have a high multidimensional poverty index concentrated in tribal districts, with limited progress in poverty reduction [33], which in turn restricts access to healthcare and immunization services. Conversely, "hotspots" were observed where districts with higher rates of 1st and 2nd birth-order children were associated with high immunization coverage, particularly in southern India, reflecting better health infrastructure and higher socioeconomic status in those regions [34]. In contrast, immunization "cold spots" emerged in many districts of Uttar Pradesh and Bihar and several north-eastern states, where similar birth-order profiles were associated with persistently low immunization coverage. This pattern likely reflects underlying demographic and socioeconomic disadvantages in these regions, compounded by both demand- and supply-side barriers, including limited awareness and acceptance of immunization services, challenges in accessing health facilities, and pervasive poverty [4]. Notably, irrespective of child, maternal, healthcare, and community characteristics, the odds of full immunization coverage drastically declined with increasing maternal birth order, which strongly indicates that maternal birth order is a pivotal determinant in shaping immunization coverage among younger offspring. However, the possible explanations are as follows. First, prior negative experiences of parents regarding adverse reactions to vaccinations of older children might make parents reluctant to vaccinate their younger children [35]. Second, parents' coincidence of their healthy, unvaccinated older children might also lead parents to perceive

vaccination as less important [36]. Third, the mother's involvement in the rearing and caring of too many offspring also compromised the scheduled doses of vaccination for the younger offspring [37]. Fourth, the parental attention and care are often greater for the first-born child compared to those of later birth orders, potentially influencing immunization completion among later-born children [14].

Moreover, multilevel logistic regression further revealed that the sex of the child, child size at birth, maternal age, maternal education, marital status, media exposure, wealth index, religion, social category, number of ANC visits, place of delivery, place of residence, and geographical region were significant factors of child immunization. In particular, the odds of immunization were 14% higher among average-sized children, which might be because mothers who deliver average-sized children are often found to be very conscious of their child's health, thereby making them regular visits to health care services such as immunization programmes. These findings were also confirmed by studies in Somalia [30], India [38], and Ethiopia [39]. Children born to mothers who were 35 or above had a 34% greater likelihood of immunization coverage than those whose mothers were aged 15–24 years, also reaffirming the studies in Ethiopia [40]and elsewhere [41]. This might be explained by the fact that older mothers are more experienced, responsible, and well-informed about the utility of immunization. In line with prior studies [30,42], our study also showed that children born to women who completed primary and secondary education had a greater likelihood of immunization coverage. A possible reason could be that educated mothers are usually subject to being aware of the implications of immunization on the one hand and have the decision-making ability to make choices for better health and well-being of their offspring, which eventually shapes positive health-seeking behaviour among them. In line with a study conducted in Ghana [43], the coverage was also found to be prevalent among children who were born to currently married women, which might be because children born outside marriage are not considered legitimate in the Indian context [44]and mothers with children born outside wedlock are considered taboo and have comparatively less psychological stability than married ones.

Notably, children born to women with partial media exposure had 28% higher odds of immunization coverage, which is consistent with the results of previous studies [41,42]. This was possible because media exposure promotes both positive implications and outreach-related information regarding health care services, such as immunization. Additionally, children belonging to economically well-off families also had a greater likelihood of being the same and were supported by previous studies [30,38]. This might be because wealthier families usually have better access to education, resources, and information, which positively shapes health-seeking behaviour. Notably, significant disparities were also observed along religious and gender lines. Muslim children had a 28% lower likelihood of immunization coverage than their Hindu counterparts (aOR: 0.72). This finding aligns with prior studies in India and may be partly due to vaccine hesitancy influenced by misinformation or a lack of trust in public health systems within some communities, as documented in sociocultural health literature. Furthermore, the analysis found that female children had slightly lower odds of being fully immunized compared to male children (aOR: 0.95). Though the effect size is modest, this finding is concerning and may reflect lingering sociocultural biases in healthcare-seeking for female children, a phenomenon documented in the broader public health literature on India. These findings aligned with prior studies in India [14,38] and were plausible due to hesitance towards immunization as induced by negative connotations apprehended by religious doctrine [45]. Notably, the likelihood of coverage was 21% greater among children who were scheduled tribe than their scheduled caste counterparts. However, it might be credited to the active role of ASHA/ANM/AWW workers in driving healthcare utilization among tribal children [46].

In addition, children born to women who underwent ≥ 4 ANC visits had a 62% greater likelihood of being immunized than those who did not. This finding is in agreement with those of studies in East Africa [41], Ethiopia [40], and Indonesia [47]. However, this was because maternal ANC visits epitomize the awareness and priorities of mothers for good health for both themselves and their younger ones on the one hand, and the exposure to health services during the gestation period also assures positive healthcare-seeking behaviour on the other. In addition, children in rural settings have 32% greater odds of full immunization coverage than their urban counterparts, as affirmed by an Indian study [38]. It might be credited to the strengthening of primary health care in rural settings on the one hand and the incentivization programmes

like Janani Suraksha Yojana, which have drastically improved institutional deliveries at public institutions. Accordingly, children born to women who delivered at a public institution had 64% higher odds of being fully immunized than those delivered at home (aOR: 1.64). Likewise, children born to women who delivered at public institutions were 64% more likely to be fully immunized than those who delivered at home. This finding is consistent with a previous study [48] and could be ascribed to maternal exposure to counselling regarding the benefits of vaccinations soon after their delivery, on the one hand, and experiences of institutional maternal services on the other, which altogether creates a positive outlook towards utilization of health care services like immunization. Moreover, children from the northeast region were 48% less likely to be fully vaccinated than their northern counterparts, which conformed to a previous study in India [49]. For example, a study in Nagaland [50] depicts that the availability of buses to the health centre is associated with a two-fold increase in the odds of coverage of BCG and OPV3. A plausible reason could be that the hilly geographical terrain, poor healthcare infrastructure, and lack of trained health personnel have made this region less susceptible to the utilization of health care services [51] which creates severe programmatic and logistical challenges for consistent vaccine delivery and health worker outreach.

### Limitations of the study

The present study had several limitations. First, it is mainly based on observational and cross-sectional analyses and identifies the association. However, they did not establish a causal relationship between birth order and immunization coverage that underscores the need for further longitudinal study. Second, the immunization data in the NFHS are partly based on maternal recall, particularly when vaccination cards are unavailable, which may introduce reporting bias. For example, mothers may claim their child received a vaccine when they did not or may forget/fail to mention a vaccine that their child actually received. Third, it mainly focused on 12–23-month-old children who received all basic vaccines at any time before the survey without accounting for the exact timing of immunization.

### Conclusions

This study highlights a significant and consistent association between birth order and full immunization coverage among children aged 12–23 months in India. The analysis revealed that children from higher birth orders (three or later) are remarkably less likely to receive complete immunization than first-born children. This may be due to possible factors, such as the negative experience of parents regarding vaccination or decreased attention toward subsequent children in larger families. Similarly, the probability of full immunization coverage among children is significantly lower among female children, belonging to Muslim religions, and residing in the northeastern part of India, particularly when they are not the firstborn.

The findings of this study underscore that designing vaccination campaigns in such a way that specifically focuses on families with three or more children. Immunization reminders should be systematically aligned with routine antenatal care (ANC), postnatal care, and child growth monitoring services to ensure repeated contact with mothers of multiple children. Furthermore, high-parity households should be prioritized in Mission Indradhanush outreach through family-centric IEC (Information, Education and Communication) campaigns, supported by the proactive, household-level engagement of ASHA workers, including door-to-door follow-ups and counselling during immunization rounds. In addition, maintaining parity-specific household lists at the sub-centre level and integrating them into micro-plans could strengthen last-mile delivery. Collectively, these programmatic strategies would promote equitable access and improve immunization coverage among children from larger families.

To achieve universal immunization goals, government and non-government organizations should come forward and implement preventive strategies that emphasize equitable access, community engagement, family-focused outreach, and awareness campaigns, particularly in households with multiple children.

## Acknowledgments

The authors express sincere gratitude to the International Institute of Population Science (IIPS), Mumbai, for making the dataset available.

## Author contributions

**Conceptualization:** Pradip Chouhan.

**Data curation:** Tanmoy Ghosh.

**Methodology:** Tanmoy Ghosh, Puja Das, Apurba Sarkar, Pradip Chouhan.

**Software:** Tanmoy Ghosh.

**Visualization:** Tanmoy Ghosh, Puja Das.

**Writing – original draft:** Tanmoy Ghosh, Puja Das, Apurba Sarkar.

**Writing – review & editing:** Tanmoy Ghosh, Puja Das, Apurba Sarkar, Pradip Chouhan.

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
