## [Decision Letter · Decision Letter 0]

18 Jul 2025

PONE-D-25-28699How does Birth Order Influence Full Immunization Coverage among Children Aged 12-23 Months in India? Evidence from the National Family Health SurveyPLOS ONE?

Dear Dr. Ghosh,

Thank you for submitting your manuscript to PLOS ONE. After careful consideration, we feel that it has merit but does not fully meet PLOS ONE’s publication criteria as it currently stands. Therefore, we invite you to submit a revised version of the manuscript that addresses the points raised during the review process.

We look forward to receiving your revised manuscript.

Kind regards,

Roy, A. Ph. D

Academic Editor

PLOS ONE

Journal Requirements:

2. In the online submission form, you indicated that your data will be submitted to a repository upon acceptance. We strongly recommend all authors deposit their data before acceptance, as the process can be lengthy and hold up publication timelines. Please note that, though access restrictions are acceptable now, your entire minimal dataset will need to be made freely accessible if your manuscript is accepted for publication. This policy applies to all data except where public deposition would breach compliance with the protocol approved by your research ethics board. If you are unable to adhere to our open data policy, please kindly revise your statement to explain your reasoning and we will seek the editor's input on an exemption.

3. We note that Figure 2 in your submission contain [map/satellite] images which may be copyrighted. All PLOS content is published under the Creative Commons Attribution License (CC BY 4.0), which means that the manuscript, images, and Supporting Information files will be freely available online, and any third party is permitted to access, download, copy, distribute, and use these materials in any way, even commercially, with proper attribution. For these reasons, we cannot publish previously copyrighted maps or satellite images created using proprietary data, such as Google software (Google Maps, Street View, and Earth). For more information, see our copyright guidelines: http://journals.plos.org/plosone/s/licenses-and-copyright.

1. You may seek permission from the original copyright holder of Figure 2 to publish the content specifically under the CC BY 4.0 license.

Reviewers' comments:

Reviewer's Responses to Questions

**Comments to the Author**

1. Is the manuscript technically sound, and do the data support the conclusions?

Reviewer #1: Yes

Reviewer #2: Partly

2. Has the statistical analysis been performed appropriately and rigorously?

Reviewer #1: Yes

Reviewer #2: No

3. Have the authors made all data underlying the findings in their manuscript fully available?

Reviewer #1: Yes

Reviewer #2: Yes

4. Is the manuscript presented in an intelligible fashion and written in standard English?

Reviewer #1: Yes

Reviewer #2: No

Reviewer #1: This article addresses an important and policy-relevant public health issue, the influence of birth order on the full immunization coverage of children. In the context of low- and middle-income countries, such as in India where immunization coverage remains uneven and child morbidity and mortality is a persistent concern, understanding how birth order shapes immunization uptake is both timely and significant. The manuscript offers valuable insights into intra-household dynamics affecting full immunization coverage. However, there are several aspects that need further clarification to enhance the overall quality and impact of the study. Specific comments and suggestions are given below

Abstract

In abstract include result value along with the statement such as “the likelihood of full immunization declines significantly with ….” Include OR throughout the abstract where applicable.

Introduction

Consider linking the study to relevant Sustainable Development Goals (SDGs) in introduction section to enhance the global significance of the study.

The manuscript would benefit from a more robust justification of the study rationale. While the topic is relevant, the authors should better establish why this specific research is needed by engaging more thoroughly with existing literature.

The stated objective of the study is “to determine how birth order influences full immunization coverage among children aged 12-23 months in India”. However, the manuscript includes spatial analysis (e.g., Moran’s I) to explore the geographic distribution or clustering of the variables, which is not reflected in the stated objectives. To maintain alignment between the study objectives and the methods employed, it is recommended that the authors revise the objective section to reflect the inclusion of spatial analysis. For instance, “Furthermore, this study also aims to explore the spatial clustering of…….”

Methodology

The manuscript presents maternal birth order as the main explanatory variable for full immunization coverage; however, it lacks clarity regarding independent and control variables (L: 100–102). I recommend that the authors explicitly define birth order as the key independent variable and clearly specify relevant control variables (e.g., sex of child (male and female) ……….) to account for potential confounders.

Discussion

The discussion on spatial association clusters and hotspots (L: 261–262) appears somewhat superficial. While Odisha and Chhattisgarh are identified as high-risk zones, the manuscript does not sufficiently explore the underlying causes. A more thorough integration of relevant secondary literature would enhance the depth of the explanation.

While the authors provide plausible explanations for the observed association between maternal birth order and immunization coverage (L: 266-271), these explanations are not supported by relevant citations. It is therefore recommended that the authors include supportive evidence from previous national and international studies to strengthen and validate their explanations.

The observed association between higher birth order and lower immunization coverage is acknowledged, yet the manuscript does not adequately discuss its policy and practical implications. I recommend that the authors briefly discuss how their findings could inform targeted policy interventions and practical strategies aimed at improving universal immunization coverage, particularly addressing the gap.

In Table 1, variables such as sex of the household head, wealth index, religion, and social category are more appropriately classified as household characteristics rather than maternal characteristics. In the multilevel analysis (Table 3, Model 3), the authors should either introduce a separate model specifically for household-level characteristics or clearly integrate both maternal and household characteristics within the same model.

Reviewer #2: Comments:

The abstract effectively captures the research question and findings, but it lacks quantitative specificity. Include effect sizes or percentage differences in full immunization between birth orders to strengthen the quantitative rigor of the abstract.

The rationale for exploring birth order as a determinant is sound, but the novelty could be better emphasized. Clarify how your study fills the knowledge gap, especially regarding spatial analysis of birth order and immunization in India. A clearer statement of contribution will highlight the study’s uniqueness.

While NFHS-5 is an appropriate and rich data source, the justification for restricting the age group to 12–23 months should be more explicit.

Add a sentence justifying why this age group is ideal (e.g., global standards for “fully immunized” status).

Spatial analysis using bivariate Moran’s I is appreciated but briefly described.Clarify spatial unit of analysis (districts) and provide rationale for using bivariate Moran’s I over other spatial correlation methods (e.g., GWR, Local MoIn the multilevel model section, the role of PSU (Primary Sampling Unit) as the level of clustering needs more explanation.State why PSU is the chosen cluster level and whether random intercept or random slope models were tested.

The logistic regression results are comprehensive, but effect size interpretations need refinement. When reporting odds ratios, clearly interpret them (e.g., “Children of 6th or higher birth order had 34% lower odds…”). Avoid merely reporting directionality without context.

Spatial LISA maps are referenced, but not all are described or discussed in detail. Provide more narrative interpretation of LISA outputs. What explains clusters in specific regions like Odisha or the Northeast? This would strengthen the spatial component.

The discussion includes a good synthesis of existing literature, but tends to be descriptive rather than analytical. Engage more critically with the findings. For example, why do children from the Northeast show lower coverage despite rural areas overall doing better? Discuss potential programmatic and logistical causes.

Religious and gender disparities are noted but treated cautiously. Be careful with sensitive interpretations but avoid generalizations. Back these with sociocultural or policy literature to reduce speculative tone.

The limitations are mentioned but need elaboration. Clarify the potential recall bias from maternal reporting and misclassification of vaccine status. Also note lack of longitudinal design limiting causal inference.

While recommendations are included, they are somewhat generic. Offer more actionable insights—e.g., “High-parity households should be prioritized under Mission Indradhanush outreach using family-centric IEC campaigns and ASHA worker engagement.

Reference formatting is inconsistent (e.g., “[6–8]” vs. full author names in some). Tables are informative but need better captions (e.g., “Model 6 includes all covariates” could be added). Consider including a robustness check (e.g., sensitivity to child sex stratification) as supplementary analysis.

**Do you want your identity to be public for this peer review?** For information about this choice, including consent withdrawal, please see our Privacy Policy

Reviewer #1: No

Reviewer #2: No

---

## [Author Response · Author response to Decision Letter 1]

14 Oct 2025

Dear Editor,

We are submitting our revised paper titled “How does Birth Order Influence Full Immunization Coverage among Children Aged 12-23 Months in India? Evidence from the National Family Health Survey”. We have made revisions to our paper again after paying close attention to the comments and suggestions by the editor and reviewers. We sincerely thank you for the valuable and constructive feedback that we have used to improve the quality of our manuscript.

Reviewer(s)' Comments:

Reviewer:1

This article addresses an important and policy-relevant public health issue, the influence of birth order on the full immunization coverage of children. In the context of low- and middle-income countries, such as in India where immunization coverage remains uneven and child morbidity and mortality is a persistent concern, understanding how birth order shapes immunization uptake is both timely and significant. The manuscript offers valuable insights into intra-household dynamics affecting full immunization coverage. However, there are several aspects that need further clarification to enhance the overall quality and impact of the study. Specific comments and suggestions are given below.

Abstract

In abstract include result value along with the statement such as “the likelihood of full immunization declines significantly with ….” Include OR throughout the abstract where applicable.

Response: Thank you for your kind appreciation. We have edited accordingly in the revised manuscript. Please see line: 33-36.

Introduction

Consider linking the study to relevant Sustainable Development Goals (SDGs) in introduction section to enhance the global significance of the study. The manuscript would benefit from a more robust justification of the study rationale. While the topic is relevant, the authors should better establish why this specific research is needed by engaging more thoroughly with existing literature.

Response: Thank you sir/madam for your suggestions. We have revised the introduction section accordingly. Please go through lines: 67-78, 81-83.

The stated objective of the study is “to determine how birth order influences full immunization coverage among children aged 12-23 months in India”. However, the manuscript includes spatial analysis (e.g., Moran’s I) to explore the geographic distribution or clustering of the variables, which is not reflected in the stated objectives. To maintain alignment between the study objectives and the methods employed, it is recommended that the authors revise the objective section to reflect the inclusion of spatial analysis. For instance, “Furthermore, this study also aims to explore the spatial clustering of…….”

Response: We have edited accordingly in revised manuscript. Please see line: 76-78.

Methodology

The manuscript presents maternal birth order as the main explanatory variable for full immunization coverage; however, it lacks clarity regarding independent and control variables (L: 100–102). I recommend that the authors explicitly define birth order as the key independent variable and clearly specify relevant control variables (e.g., sex of child (male and female) ……….) to account for potential confounders.

Response: We have edited accordingly in the revised manuscript. Please see the line: 116-118.

Discussion

The discussion on spatial association clusters and hotspots (L: 261–262) appears somewhat superficial. While Odisha and Chhattisgarh are identified as high-risk zones, the manuscript does not sufficiently explore the underlying causes. A more thorough integration of relevant secondary literature would enhance the depth of the explanation. While the authors provide plausible explanations for the observed association between maternal birth order and immunization coverage (L: 266-271), these explanations are not supported by relevant citations. It is therefore recommended that the authors include supportive evidence from previous national and international studies to strengthen and validate their explanations. Response: Thank you, sir/madam, for your suggestions. We have revised the discussion section accordingly. Please go through lines: 297-302.

The observed association between higher birth order and lower immunization coverage is acknowledged, yet the manuscript does not adequately discuss its policy and practical implications. I recommend that the authors briefly discuss how their findings could inform targeted policy interventions and practical strategies aimed at improving universal immunization coverage, particularly addressing the gap.

Response: We have revised policy implications accordingly. Kindly go through lines: 398-404.

In Table 1, variables such as sex of the household head, wealth index, religion, and social category are more appropriately classified as household characteristics rather than maternal characteristics. In the multilevel analysis (Table 3, Model 3), the authors should either introduce a separate model specifically for household-level characteristics or clearly integrate both maternal and household characteristics within the same model.

Response: We have edited accordingly in the revised manuscript. Kindly go through revised tables 1, 2 and 3.

Reviewer 2

The abstract effectively captures the research question and findings, but it lacks quantitative specificity. Include effect sizes or percentage differences in full immunization between birth orders to strengthen the quantitative rigor of the abstract.

Response: Thank you sir/madam for your suggestion. We have edited accordingly in the revised manuscript. Kindly go through lines: 33-36.

The rationale for exploring birth order as a determinant is sound, but the novelty could be better emphasized. Clarify how your study fills the knowledge gap, especially regarding spatial analysis of birth order and immunization in India.

Response: Thank you for your kind appreciation. We have revised accordingly. Please go through line: 67-74, 76-77.

A clearer statement of contribution will highlight the study’s uniqueness. While NFHS-5 is an appropriate and rich data source, the justification for restricting the age group to 12–23 months should be more explicit. Add a sentence justifying why this age group is ideal (e.g., global standards for “fully immunized” status).

Response: Thank you sir/madam for your suggestion. We have edited accordingly in the revised manuscript. Kindly go through lines: 106-109.

Spatial analysis using bivariate Moran’s I is appreciated but briefly described. Clarify spatial unit of analysis (districts) and provide rationale for using bivariate Moran’s I over other spatial correlation methods (e.g., GWR, Local MoIn the multilevel model section, the role of PSU (Primary Sampling Unit) as the level of clustering needs more explanation. State why PSU is the chosen cluster level and whether random intercept or random slope models were tested.

Response: We have incorporated your recommendations accordingly. Please go through the statistical analysis section line: 137-145, 156-162.

The logistic regression results are comprehensive, but effect size interpretations need refinement. When reporting odds ratios, clearly interpret them (e.g., “Children of 6th or higher birth order had 34% lower odds…”). Avoid merely reporting directionality without context.

Response: We have elaborated the interpretation section. Kindly go through the lines: 245-249.

Spatial LISA maps are referenced, but not all are described or discussed in detail. Provide more narrative interpretation of LISA outputs. What explains clusters in specific regions like Odisha or the Northeast? This would strengthen the spatial component.

Response: Thank you for your valuable suggestion. We have elaborated accordingly in the revised manuscript. Kindly go through the lines: 297-302.

The discussion includes a good synthesis of existing literature, but tends to be descriptive rather than analytical. Engage more critically with the findings. For example, why do children from the Northeast show lower coverage despite rural areas overall doing better? Discuss potential programmatic and logistical causes.

Response: We have incorporated your recommendations accordingly. Please go through the discussion section line: 376-378.

Religious and gender disparities are noted but treated cautiously. Be careful with sensitive interpretations but avoid generalisations. Back these with sociocultural or policy literature to reduce speculative tone.

Response: Thank you for your suggestion.

The limitations are mentioned but need elaboration. Clarify the potential recall bias from maternal reporting and misclassification of vaccine status. Also note the lack of longitudinal design limiting causal inference.

Response: We have edited accordingly. Kindly go through line: 381-383, 384-386.

While recommendations are included, they are somewhat generic. Offer more actionable insights—e.g., “High-parity households should be prioritized under Mission Indradhanush outreach using family-centric IEC campaigns and ASHA worker engagement.

Response: Thank you for your suggestions. We have incorporated your recommendations accordingly. Please go through the conclusion section line: 398-404.

Reference formatting is inconsistent (e.g., “[6–8]” vs. full author names in some).

Response: We have revised the references as per journal guidelines.

Tables are informative but need better captions (e.g., “Model 6 includes all covariates” could be added).

Response: We have edited accordingly. Kindly go through line: 165-166.

Consider including a robustness check (e.g., sensitivity to child sex stratification) as supplementary analysis.

Response: Thank you for your insightful observation. Since our study primarily focuses upon the interplay between maternal birth order and child immunization status through the spatial lens, we have not accommodated sensitivity analysis. However, in our future study, we would definitely incorporate the same.

---

## [Decision Letter · Decision Letter 1]

17 Nov 2025

PONE-D-25-28699R1How does Birth Order Influence Full Immunization Coverage among Children Aged 12-23 Months in India? Evidence from the National Family Health SurveyPLOS ONE?

Dear Dr. Ghosh,

Thank you for submitting your manuscript to PLOS ONE. After careful consideration, we feel that it has merit but does not fully meet PLOS ONE’s publication criteria as it currently stands. Therefore, we invite you to submit a revised version of the manuscript that addresses the points raised during the review process.

We look forward to receiving your revised manuscript.

Kind regards,

Avijit Roy, Ph. D

Academic Editor

PLOS ONE

Journal Requirements:

Reviewers' comments:

Reviewer's Responses to Questions

**Comments to the Author**

Reviewer #1: All comments have been addressed

Reviewer #2: (No Response)

2. Is the manuscript technically sound, and do the data support the conclusions?

Reviewer #1: Yes

Reviewer #2: Yes

3. Has the statistical analysis been performed appropriately and rigorously?

Reviewer #1: Yes

Reviewer #2: No

4. Have the authors made all data underlying the findings in their manuscript fully available?

Reviewer #1: Yes

Reviewer #2: (No Response)

5. Is the manuscript presented in an intelligible fashion and written in standard English?

Reviewer #1: Yes

Reviewer #2: No

Reviewer #1: I have carefully evaluated the revised version and I am pleased to note that the authors have satisfactorily addressed all the previous comments and suggestions. The manuscript is now clearly structured, the methodology is well-explained, and the results are presented coherently. The revisions have improved both the clarity and the academic quality of the paper.

I find no further issues that require modification. Therefore, I recommend the manuscript for acceptance in its current form.

Reviewer #2: Upon reviewing the resubmitted manuscript and the accompanying response document, it is noted that although the authors have provided written replies to each of the reviewer comments, the manuscript itself does not appear to have been revised accordingly. In several instances, the authors state that they have “addressed” the concern or “considered” the suggestion; however, the corresponding modifications are not reflected in the main text, tables, figures, or reference list. As a result, the revisions remain theoretical rather than substantive, and the manuscript in its current form does not adequately incorporate or demonstrate the improvements claimed by the authors in their response.

It is standard scholarly practice that when responding to reviewer comments, authors not only explain how each suggestion was handled but also ensure that the manuscript is visibly and systematically revised to reflect these changes. Simply providing written explanations without updating the manuscript limits the transparency of the revision process and does not allow reviewers or editors to verify whether the issues raised have been meaningfully resolved. The purpose of the revision stage is to enhance the clarity, methodological rigor, coherence, and academic contribution of the paper based on constructive peer feedback. Therefore, the absence of corresponding textual or structural changes in the manuscript is a matter of concern.

The authors are kindly advised to carefully review all comments once more and revise the manuscript in a thorough and detailed manner. Each claimed adjustment should be directly traceable in the revised text. To facilitate the review process, the authors are strongly encouraged to provide:

1. A tracked-changes version of the manuscript, clearly highlighting all modifications; and

2. A clean version of the revised manuscript.

These steps will allow the reviewers and editorial board to efficiently verify the revisions and assess the extent to which the manuscript has improved. Until the suggested changes are fully implemented in the manuscript itself, it cannot be considered satisfactorily revised.

**Do you want your identity to be public for this peer review?** For information about this choice, including consent withdrawal, please see our Privacy Policy

Reviewer #1: No

Reviewer #2: No

---

## [Author Response · Author response to Decision Letter 2]

3 Jan 2026

Dear Editor,

We are submitting our revised paper titled “How does Birth Order Influence Full Immunization Coverage among Children Aged 12-23 Months in India? Evidence from the National Family Health Survey”. We have made revisions to our paper again after paying close attention to the comments and suggestions by the editor and reviewers. We sincerely thank you for the valuable and constructive feedback that we have used to improve the quality of our manuscript.

Reviewer(s)' Comments:

Reviewer:1

I have carefully evaluated the revised version and I am pleased to note that the authors have satisfactorily addressed all the previous comments and suggestions. The manuscript is now clearly structured, the methodology is well-explained, and the results are presented coherently. The revisions have improved both the clarity and the academic quality of the paper.

I find no further issues that require modification. Therefore, I recommend the manuscript for acceptance in its current form.

Response: Thank you, sir/madam, for your valuable feedback. We appreciate the recommendation for acceptance and thank the reviewer for their valuable input throughout the review process.

Reviewer 2

Upon reviewing the resubmitted manuscript and the accompanying response document, it is noted that although the authors have provided written replies to each of the reviewer comments, the manuscript itself does not appear to have been revised accordingly. In several instances, the authors state that they have “addressed” the concern or “considered” the suggestion; however, the corresponding modifications are not reflected in the main text, tables, figures, or reference list. As a result, the revisions remain theoretical rather than substantive, and the manuscript in its current form does not adequately incorporate or demonstrate the improvements claimed by the authors in their response.

It is standard scholarly practice that when responding to reviewer comments, authors not only explain how each suggestion was handled but also ensure that the manuscript is visibly and systematically revised to reflect these changes. Simply providing written explanations without updating the manuscript limits the transparency of the revision process and does not allow reviewers or editors to verify whether the issues raised have been meaningfully resolved. The purpose of the revision stage is to enhance the clarity, methodological rigor, coherence, and academic contribution of the paper based on constructive peer feedback. Therefore, the absence of corresponding textual or structural changes in the manuscript is a matter of concern.

The authors are kindly advised to carefully review all comments once more and revise the manuscript in a thorough and detailed manner. Each claimed adjustment should be directly traceable in the revised text. To facilitate the review process, the authors are strongly encouraged to provide:

1. A tracked-changes version of the manuscript, clearly highlighting all modifications; and

2. A clean version of the revised manuscript.

These steps will allow the reviewers and editorial board to efficiently verify the revisions and assess the extent to which the manuscript has improved. Until the suggested changes are fully implemented in the manuscript itself, it cannot be considered satisfactorily revised.

Response: We sincerely thank the reviewer for the careful reassessment of the revised manuscript and for highlighting this important concern. We have carefully re-examined all reviewer comments and have thoroughly and systematically revised the manuscript to address them in detail.

The abstract effectively captures the research question and findings, but it lacks quantitative specificity. Include effect sizes or percentage differences in full immunization between birth orders to strengthen the quantitative rigor of the abstract.

Response: Thank you, sir/madam, for your suggestion. We have edited accordingly in the revised manuscript. Kindly go through lines: 33-36.

The rationale for exploring birth order as a determinant is sound, but the novelty could be better emphasized. Clarify how your study fills the knowledge gap, especially regarding spatial analysis of birth order and immunization in India.

Response: Thank you for your kind appreciation. We have revised accordingly. Please go through line: 67-74, 76-77, 81-83.

A clearer statement of contribution will highlight the study’s uniqueness. While NFHS-5 is an appropriate and rich data source, the justification for restricting the age group to 12–23 months should be more explicit. Add a sentence justifying why this age group is ideal (e.g., global standards for “fully immunized” status).

Response: Thank you sir/madam for your suggestion. We have edited accordingly in the revised manuscript. Kindly go through lines: 106-109.

Spatial analysis using bivariate Moran’s I is appreciated but briefly described. Clarify spatial unit of analysis (districts) and provide rationale for using bivariate Moran’s I over other spatial correlation methods (e.g., GWR, Local MoIn the multilevel model section, the role of PSU (Primary Sampling Unit) as the level of clustering needs more explanation. State why PSU is the chosen cluster level and whether random intercept or random slope models were tested.

Response: We have incorporated your recommendations accordingly. Please go through the statistical analysis section line: 137-145, 156-162.

The logistic regression results are comprehensive, but effect size interpretations need refinement. When reporting odds ratios, clearly interpret them (e.g., “Children of 6th or higher birth order had 34% lower odds…”). Avoid merely reporting directionality without context.

Response: We have elaborated the interpretation section. Kindly go through the lines: 245-249.

Spatial LISA maps are referenced, but not all are described or discussed in detail. Provide more narrative interpretation of LISA outputs. What explains clusters in specific regions like Odisha or the Northeast? This would strengthen the spatial component.

Response: Thank you for your valuable suggestion. We have elaborated accordingly in the revised manuscript. Kindly go through the lines: 294-308.

The discussion includes a good synthesis of existing literature, but tends to be descriptive rather than analytical. Engage more critically with the findings. For example, why do children from the Northeast show lower coverage despite rural areas overall doing better? Discuss potential programmatic and logistical causes.

Response: We have incorporated your recommendations accordingly. Please go through the discussion section line: 380-385.

Religious and gender disparities are noted but treated cautiously. Be careful with sensitive interpretations but avoid generalisations. Back these with sociocultural or policy literature to reduce speculative tone.

Response: Thank you for your suggestion. Kindly go through line: 348-358.

The limitations are mentioned but need elaboration. Clarify the potential recall bias from maternal reporting and misclassification of vaccine status. Also note the lack of longitudinal design limiting causal inference.

Response: We have edited accordingly. Kindly go through line: 388-393.

While recommendations are included, they are somewhat generic. Offer more actionable insights—e.g., “High-parity households should be prioritized under Mission Indradhanush outreach using family-centric IEC campaigns and ASHA worker engagement.

Response: Thank you for your suggestions. We have incorporated your recommendations accordingly. Please go through the conclusion section line: 405-415.

Reference formatting is inconsistent (e.g., “[6–8]” vs. full author names in some).

Response: We have revised the references as per journal guidelines.

Tables are informative but need better captions (e.g., “Model 6 includes all covariates” could be added).

Response: We have edited accordingly. Kindly go through line: 165-166.

Consider including a robustness check (e.g., sensitivity to child sex stratification) as supplementary analysis.

Response: Thank you for your insightful observation. Since our study primarily focuses upon the interplay between maternal birth order and child immunization status through the spatial lens, we have not accommodated sensitivity analysis. However, in our future study, we would definitely incorporate the same.

---

## [Decision Letter · Decision Letter 2]

20 Jan 2026

How does Birth Order Influence Full Immunization Coverage among Children Aged 12-23 Months in India? Evidence from the National Family Health Survey

PONE-D-25-28699R2

Dear Mr. Ghosh,

We’re pleased to inform you that your manuscript has been judged scientifically suitable for publication and will be formally accepted for publication once it meets all outstanding technical requirements.

Kind regards,

Avijit Roy, Ph. D

Academic Editor

PLOS One

Additional Editor Comments (optional):

Reviewers' comments:

Reviewer's Responses to Questions

**Comments to the Author**

Reviewer #2: All comments have been addressed

2. Is the manuscript technically sound, and do the data support the conclusions?

Reviewer #2: Yes

3. Has the statistical analysis been performed appropriately and rigorously?

Reviewer #2: Yes

4. Have the authors made all data underlying the findings in their manuscript fully available?

Reviewer #2: Yes

5. Is the manuscript presented in an intelligible fashion and written in standard English?

Reviewer #2: Yes

Reviewer #2: I have carefully reviewed the revised version of the manuscript entitled “How does Birth Order Influence Full Immunization Coverage among Children Aged 12–23 Months in India? Evidence from the National Family Health Survey” (Manuscript ID: PONE-D-25-28699R2).

I am pleased to note that the authors have satisfactorily addressed all the comments and suggestions raised during the previous rounds of review. The manuscript is now clearly structured, the objectives are well articulated, the methodology is appropriately justified, and the results are presented in a coherent and transparent manner. The revisions have significantly improved the clarity, rigor, and overall academic quality of the paper.

In its current form, I find no further issues that require clarification or modification. The study makes a meaningful contribution to the literature on child immunization and public health in India, particularly by highlighting the role of birth order using robust statistical and spatial analytical approaches.

Therefore, I recommend that the manuscript be accepted for publication in its present form.

**Do you want your identity to be public for this peer review?** For information about this choice, including consent withdrawal, please see our Privacy Policy

Reviewer #2: No

---

## [Editor Report · Acceptance letter]

PONE-D-25-28699R2

PLOS One

Dear Dr. Ghosh,

I'm pleased to inform you that your manuscript has been deemed suitable for publication in PLOS One. Congratulations! Your manuscript is now being handed over to our production team.

Kind regards,

on behalf of

Dr. Avijit Roy

Academic Editor

PLOS One